# Goal-Auxiliary Actor-Critic for 6D Robotic Grasping with Point Clouds

**Lirui Wang[1], Yu Xiang[2], Wei Yang[2], Arsalan Mousavian[2], Dieter Fox[1,2]**
[1]University of Washington, [2]NVIDIA
liruiw@cs.washington.edu,{yux, weiy, amousavian, dieterf}@nvidia.com

**Abstract:** 6D robotic grasping beyond top-down bin-picking scenarios is a challenging task. Previous solutions based on 6D grasp synthesis with robot motion planning usually operate in an open-loop setting, which are sensitive to grasp synthesis errors. In this work, we propose a new method for learning closed-loop control policies for 6D grasping. Our policy takes a segmented point cloud of an object from an egocentric camera as input, and outputs continuous 6D control actions of the robot gripper for grasping the object. We combine imitation learning and reinforcement learning and introduce a goal-auxiliary actor-critic algorithm for policy learning. We demonstrate that our learned policy can be integrated into a tabletop 6D grasping system and a human-robot handover system to improve the grasping performance of unseen objects. Videos and code are available at https://sites.google.com/view/gaddpg.

**Keywords:** 6D Robotic Grasping, Imitation Learning, Reinforcement Learning

## 1 Introduction

Robotic grasping of arbitrary objects is a challenging task. A robot needs to deal with objects it has never seen before, and generates a motion trajectory to grasp an object. Due to the complexity of the problem, majority works in the literature focus on bin-picking tasks, where top-down grasping is sufficient to pick up an object. Both grasp detection approaches [1, 2, 3] and reinforcement learning-based methods [4, 5] are introduced to tackle the top-down grasping problem. However, it is difficult for these methods to grasp objects in environments where 6D grasping is necessary, i.e., 3D translation and 3D rotation of the robot gripper, such as a cereal box on a tabletop or in a cabinet.

While 6D grasp synthesis has been studied using 3D models of objects [6, 7] and partial observations [8, 9, 10], these methods only generate 6D grasp poses of the robot gripper for an object, instead of generating a trajectory to reach and grasp the object. As a result, a motion planner is needed to plan the grasping motion according to the grasp poses. Usually, the planned trajectory is executed in an open-loop fashion since re-planning is expensive, and perception feedback during grasping as well as dynamics and contacts of the object are often ignored, which makes the grasping sensitive to grasp synthesis errors.

In this work, to overcome the limitations of open-loop 6D grasping based on grasp synthesis, we introduce a new method for learning closed-loop 6D grasping polices from partially-observed point clouds of objects (Fig. 1). Our policy takes a segmented point cloud of an object as input and directly outputs the control action of the robot gripper, which is the relative 6D pose transformation of the gripper. Our policy can be combined with different perception methods for object segmentation.

In order to learn the policy, we combine Imitation Learning (IL) and Reinforcement Learning (RL). Because the chance of lifting an object in 6D grasping is very rare if only RL is used for exploration. We obtain demonstrations using a joint motion and grasp planner [11] in simulation. Consequently, we can efficiently obtain a large number of 6D grasping trajectories using ShapeNet objects [12] with the planner. Then we learn the grasping policy based on the Deep Deterministic Policy Gradient (DDPG) algorithm [13], which is an actor-critic algorithm in RL that can utilize off-policy data from demonstrations. More importantly, we introduce a goal prediction auxiliary task to improve the policy learning, where the actor and the critic in DDPG are trained to predict the final 6D grasping

5th Conference on Robot Learning (CoRL 2021), London, UK.

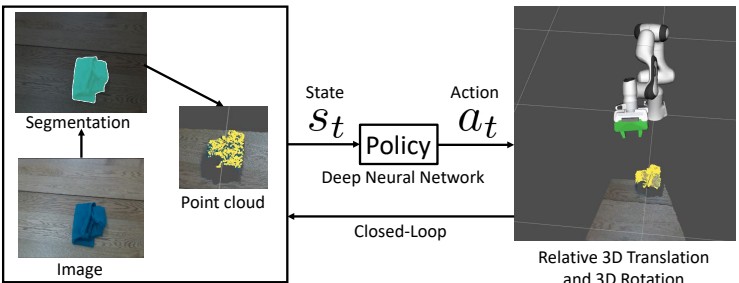

Figure 1: Illustration of our learned policy for 6D grasping. The state representation is based on the segmented point cloud of an object. The action is the relative 6D pose transformation of the robot gripper. The green gripper in the figure indicates the next target location of the gripper.

pose of the robot gripper as well. The supervision on goal prediction comes from the expert planner for objects with known 3D shape and pose. For unknown objects without 3D models available, we can still obtain the grasping goals from successful grasping rollouts of the policy, i.e., hindsight goals. This property enables our learned policy to be fine-tuned on unknown objects, which is critical for continual learning in the real world. We see that adding the auxiliary losses stabilizes training and improves the learned policy.

Overall, our contributions are: 1) We introduce a Goal-Auxiliary DDPG (GA-DDPG) algorithm for joint imitation and reinforcement learning of 6D robotic grasping. 2) We demonstrate that our learned policy can be integrated into a tabletop 6D grasping system to improve grasping performance. 3) We demonstrate that our closed-loop policy can improve human-to-robot handover.

## 2    Related Work

**Vision-based Robotic Grasping.** Grasp synthesis can be used in a planning and control pipeline for robotic grasping [2, 3, 14, 15]. Alternatively, end-to-end policy learning methods [16, 4, 17] make use of large-scale data to learn closed-loop vision-based grasping. Although RGB images are widely used as the state representation, it requires the policy to infer 3D information from 2D images. Recently, depth and segmentation masks [3], shape completion [9], deep visual features [18], keypoints [19], point clouds [20], multiple cameras [21], and egocentric views [22, 23] have been considered to improve the state representation. Motivated by these methods, we utilize point clouds of objects as our state representation, which transfers well from simulation to the real world.

**Combining Imitation Learning and Reinforcement Learning.** Model-free RL [4, 5] requires a large number of interactions even with full-state information. Therefore, many works have proposed to combine imitation learning [24] in RL. [25] augments policy gradient update with demonstration data to circumvent reward shaping and the exploration challenge. [26] uses inverse RL to learn dexterous manipulation tasks with a few human demonstrations in simulation. The closest related works to ours are [27, 28] that utilize demonstration data with off-policy RL. Despite the focus on different tasks, the main difference is that our demonstrations come from an expert planner instead of human demonstrators. Thus, we can obtain a large number of consistent demonstrations and query the expert planner during training to provide supervision on the on-policy data.

**Goals and Auxiliary Tasks in Policy Learning.** Goals are often used as extra signals to guide policy learning [29]. For goal-conditioned policies, [30, 31, 32] make the observation that every trajectory is a successful demonstration of the goal state that it reaches, thereby these methods re-label goals in rollout trajectories for effective learning. However, goals still need to be provided in testing. In 6D grasping, estimating the grasping goals is a challenging problem. On the other hand, auxiliary tasks have been used to improve RL as well [33, 34, 35, 36]. We utilize the grasping goal prediction as an auxiliary task that requires the policy to predict how to grasp the target object.

## 3    Learning 6D Grasping Policies

Our goal is to learn a closed-loop policy for 6D robotic grasping. We first introduce related background knowledge and then present our method for learning the policy.

## 3.1 Background

**Markov Decision Process.** A Markov Decision Process (MDP) is defined using the tuple: $\mathcal{M} = \{\mathcal{S}, \mathcal{R}, \mathcal{A}, \mathcal{O}, \mathcal{P}, \rho_0, \gamma\}$. $\mathcal{S}$, $\mathcal{A}$, and $\mathcal{O}$ represent the state, action, and observation space. $\mathcal{R} : \mathcal{S} \times \mathcal{A} \to \mathbb{R}$ is the reward function. $\mathcal{P} : \mathcal{S} \times \mathcal{A} \to \mathcal{S}$ is the transition dynamics. $\rho_0$ is the probability distribution over initial states and $\gamma = [0, 1)$ is a discount factor. Let $\pi : \mathcal{S} \to \mathcal{A}$ be a policy which maps states to actions. In the partially observable case, at each time $t$, the policy maps a partial observation $o_t$ of the environment to an action $a_t = \pi(o_t)$. Our goal is to learn a policy that maximizes the expected cumulative rewards $\mathbb{E}_\pi[\sum_{t=0}^{\infty} \gamma^t r_t]$, where $r_t$ is the reward at time $t$. The Q-function of the policy for a state-action pair is $Q(s, a) = \mathcal{R}(s, a) + \gamma \mathbb{E}_{s', \pi}[\sum_{t=0}^{\infty} \gamma^t r_t | s_0 = s']$, where $s'$ represents the next state of taking action $a$ in state $s$ according to the transition dynamics.

**Deep Deterministic Policy Gradient.** DDPG [13] is an actor-critic algorithm that uses off-policy data, deterministic policy gradient, and temporal difference learning. It has successful applications in continuous control [37, 27]. Specifically, the actor in DDPG learns the policy $\pi_\theta(s)$, while the critic approximates the Q-function $Q_\phi(s, a)$, where $\theta$ and $\phi$ denote the parameters of the actor and the critic, respectively. A replay buffer of transitions $\mathcal{D} = \{(s, a, r, s')\}$ is maintained during training, and examples sampled from it are used to optimize the actor and critic alternately. DDPG minimizes the following Bellman error with respect to $\phi$ :

$$\mathbb{E}_{(s,a,r,s') \sim \mathcal{D}} \left[ \frac{1}{2} \Big( Q_\phi(s, a) - \big( r + \gamma Q_\phi(s', \pi_\theta(s')) \big) \Big)^2 \right]. \tag{1}$$

Then the deterministic policy $\pi_\theta$ is trained to maximize the learned Q-function with $\max_\theta \mathbb{E}_{s \sim \mathcal{D}}(Q_\phi(s, \pi_\theta(s)))$, which resembles a policy evaluation and improvement schedule.

## 3.2 6D Grasping Policy From Point Clouds

We consider the task of grasping an arbitrary object in a closed-loop manner. Our goal is to learn a 6D grasping policy $\pi : s_t \to a_t$ that maps the state $s_t$ at time $t$ to an action $a_t$. To simplify the setting, we still use $s$ to represent the input of the policy even though observations are used. At time $t$, the action $a_t$ is parametrized by the relative 3D translation and the 3D rotation of the robot end-effector. Therefore, our learned visuomotor policy represents an operational space controller for 6D grasping. We utilize 3D point clouds of objects to represent the states, which can be computed using depth images and foreground masks of the objects. These points are transformed from the camera frame to the robot end-effector frame to represent the state. Different segmentation methods can be used to obtain a foreground mask of a target object. Our learned policy is robust to noises in the segmentation. For grasping static objects, we aggregate the point clouds from previous time steps as described in Appendix Section 6.2. For dynamic objects, no aggregation is used.

## 3.3 Demonstrations from a Joint Motion and Grasp Planner

To obtain demonstrations for 6D grasping, we utilize the Optimization-based Motion and Grasp Planner (OMG Planner) [11] as our expert. Given a planning scene and a set of pre-defined grasps on a target object from a grasp planner [6, 7], the OMG Planner plans a trajectory of the robot to grasp the object. Depending on the initial configuration of the robot, it selects different grasps as the goal. This property enables us to learn a policy that grasps objects in different ways according to the initial pose of the robot gripper.

Let $\xi = (\mathcal{T}_0, \mathcal{T}_1, \dots, \mathcal{T}_T)$ be a trajectory of the robot gripper pose generated from the OMG planner to grasp an object, where $\mathcal{T}_t \in \mathbb{SE}(3)$ is the gripper pose in the robot base frame at time $t$. Then the expert action at time $t$ can be computed as $a_t^* = \mathcal{T}_{t+1} \mathcal{T}_t^{-1}$, which is the relative transformation of the gripper between $t$ and $t + 1$. Furthermore, $\mathcal{T}_T$ from the planner is a successful grasp pose of the object. We define the expert goal at time $t$ as $g_t^* = \mathcal{T}_T \mathcal{T}_t^{-1}$, which is the relative transformation between the current gripper pose and the final grasp. Finally, the state $s_t$ can be obtained by computing the point cloud as in Section 3.2. In this way, we construct an offline dataset of demonstrations from the planner as $\mathcal{D}_{\text{expert}} = \{(s_t, a_t^*, g_t^*, s_t')\}$, where $s_t'$ is the next state.

## 3.4 Behavior Cloning and DAGGER

Using the demonstration dataset $\mathcal{D}_{\text{expert}}$, we can already train a policy network $\pi_\theta$ by applying Behavior Cloning (BC) to predict an action from a state. To minimize the distance with expert actions,

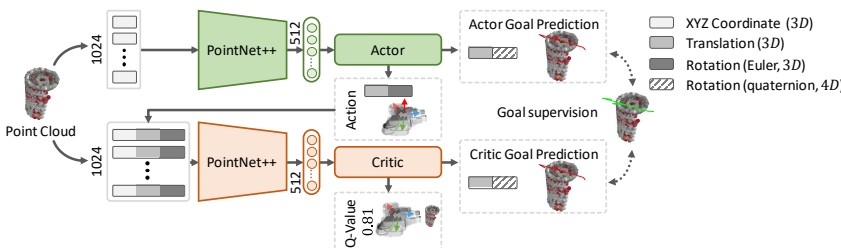

Figure 2: Our network architecture uses PointNet++ [40] for feature extraction. Both the actor and the critic are regularized to predict grasping goals as auxiliary tasks.

different loss functions on $\mathbb{SE}(3)$ can be used. In particular, we adopt the point matching loss function as defined in [38] to jointly optimize translation and rotation:

$$L_{\text{POSE}}(\mathcal{T}_1, \mathcal{T}_2) = \frac{1}{|X_g|} \sum_{x \in X_g} \|\mathcal{T}_1(x) - \mathcal{T}_2(x)\|_1, \tag{2}$$

where $\mathcal{T}_1, \mathcal{T}_2 \in \mathbb{SE}(3)$, $X_g$ denotes a set of pre-defined 3D points on the robot gripper, and L1 norm is used. Let $a_\theta = \pi_\theta(s)$ be an action predicted from the policy and $a^* = \pi^*(s)$ be an action from the expert, the behavior cloning loss is $L_{\text{BC}}(a^*, a_\theta) = L_{\text{POSE}}(a^*, a_\theta)$.

We apply DAGGER [39] to augment supervisions for rollouts from the learner. Given a sampled initial state $s_0$, the current policy $\pi_\theta$ can roll out a trajectory of states $s_0, ..., s_T$ and actions $a_0, ..., a_T$ with end-effector poses $\hat{\mathcal{T}}_0, ..., \hat{\mathcal{T}}_T$. The expert then treats each state as the initial state and generates a trajectory to grasp the object $\xi(s_t) = (\hat{\mathcal{T}}_t, \mathcal{T}_{t+1}, ..., \mathcal{T}_T)$. The first action in the plan $a_t^* = \pi^*(s_t) = \mathcal{T}_{t+1}\hat{\mathcal{T}}_t^{-1}$ is used as supervision to correct deviations of the learner from $\xi(s_t)$. A corresponding expert goal $g_t^* = \mathcal{T}_T\hat{\mathcal{T}}_t^{-1}$ can also be obtained for state $s_t$. These action and goal labels are used to construct a dataset $\mathcal{D}_{\text{dagger}} = \{(s_t, a_t, a_t^*, g_t^*, s_t')\}$.

### 3.5 Goal-Auxiliary DDPG

To handle contact-rich scenarios that are rare in the expert trajectories, we further improve the policy learning by combining RL with expert demonstrations. In our RL setting, each episode ends when the agent attempts a grasp or reaches a maximum horizon $T$. The sparse reward $r$ is an indicator function given at the end of the episode denoting if the target object is lifted or not.

Similar to DAGGER, we collect on-policy data for DDPG training. The only difference is that we do not have expert demonstration for the action $a_t^*$ and goal $g_t^*$ for a state $s_t$ from the learner policy. Instead, we first find the nearest goal from a goal set $\mathcal{G}$ by $\widetilde{g}_t = \arg\min_{g \in \mathcal{G}} L_{\text{POSE}}(g, \mathcal{T}_t)$, where $\mathcal{T}_t$ is the pose of the robot gripper at time $t$. Then the heuristic goal for $s_t$ is $g_t = \widetilde{g}_t\mathcal{T}_t^{-1}$. Note that the goal set $\mathcal{G}$ is only available if we have the 3D shape and pose of the target object. Finally, we construct a dataset for DDPG $\mathcal{D}_{\text{ddpg}} = \{(s_t, a_t, g_t, r_t, s_t')\}$, where $r_t$ is the reward at state $s_t$ and $s_t'$ is the next state of $s_t$ from the on-policy rollout. The replay buffer of DDPG training is $\mathcal{D} = \mathcal{D}_{\text{expert}} \cup \mathcal{D}_{\text{dagger}} \cup \mathcal{D}_{\text{ddpg}}$, where we augment $\mathcal{D}_{\text{expert}}$ and $\mathcal{D}_{\text{dagger}}$ with the sparse reward.

Our network architecture for training the actor-critic is shown in Fig. 2. We introduce two auxiliary tasks for the actor-critic to predict goals, which help to learn the policy and the Q-function by adding goal supervision. Given a data sample $(s, a, g, r, s')$ from the replay buffer, the critic loss is

$$L_\phi = \frac{1}{2}(Q_\phi(s, a) - y)^2 + L_{\text{AUX}}(g, g_\phi), \tag{3}$$

where $y = r + \gamma Q_{\phi'}\big(s', \pi_{\theta'}(s') + \epsilon\big)$ is the Bellman target and $g_\phi$ is the predicted goal. $Q_{\phi'}$ and $\pi_{\theta'}$ are the target networks and $\epsilon$ is a pre-defined clipped noise as in TD3 [41]. The auxiliary loss is $L_{\text{AUX}}(g, g_\phi) = L_{\text{POSE}}(g, g_\phi)$ for measuring pose errors in goal prediction.

Given a data sample $(s, a^*, g)$ from the replay buffer, the loss function for the actor is defined as

$$L_\theta = \lambda L_{\text{BC}}(a^*, a_\theta) + (1 - \lambda)L_{\text{DDPG}}(s, a_\theta) + L_{\text{AUX}}(g, g_\theta), \tag{4}$$

where $a_\theta$ and $g_\theta$ are the action and the goal predicted from the actor, respectively, and $\lambda$ is a weighting hyper-parameter to balance the losses from the expert and from the learned value function. The BC loss $L_{\text{BC}}(a^*, a_\theta) = L_{\text{POSE}}(a^*, a_\theta)$ as defined before, which prevents the learner

from moving too far away from the expert policy. The deterministic policy loss is defined as $L_{\text{DDPG}}(s, a_\theta) = -Q_\phi(s, a_\theta)$ in order to maximize Q values. Note that the expert action $a^*$ is only available if the data sample is from $\mathcal{D}_{\text{expert}}$ or $\mathcal{D}_{\text{dagger}}$. Otherwise, we do not include the BC loss in Eq. (4). Even though our method can be used with any off-policy actor-critic algorithms, in practice, we choose TD3 for its improved performance over vanilla DDPG.

## 3.6 Hindsight Goals For Fine-tuning on Unknown Objects

For unknown objects without 3D models, we cannot obtain neither expert demonstrations nor grasping goals from a pre-defined goal set. We can use hindsight goals to fine-tune our policy on unknown objects. Based on the current policy $\pi_\theta$, each learner episode rolls out a trajectory of state-actions with end-effector poses $\hat{\mathcal{T}}_0, ..., \hat{\mathcal{T}}_T$. If the grasp in the episode is successful, we know that $\hat{\mathcal{T}}_T$ is a success grasp for the target object, which is also known as the hindsight goal. Then we can compute the grasp goal for state $s_t$ in the episode as $\hat{g}_t = \hat{\mathcal{T}}_T \hat{\mathcal{T}}_t^{-1}$, which is used to supervise the actor and the critic via the goal auxiliary tasks. By using hindsight goals, we construct a dataset using on-policy rollouts on unknown objects $\mathcal{D}_{\text{hindsight}} = \{(s_t, a_t, \hat{g}_t, r_t, s'_t)\}$, where $\hat{g}_t$ is the hindsight goal. We finetune the pretrained actor-critic network on the dataset $\mathcal{D} = \mathcal{D}_{\text{expert}} \cup \mathcal{D}_{\text{hindsight}}$. The algorithm for training with hindsight goals can be found in Appendix Section 6.1.

# 4 Experiments

**Simulation Environment.** We experiment with the Franka Emika Panda arm, a 7-DOF arm with a parallel gripper. We use ShapeNet [12] and YCB objects [42] as our object repository. A task scene is generated by dropping a sampled target object with a random pose on a table in the PyBullet Simulator [43]. The maximum horizon for the policy is $T = 30$. An episode is terminated once a grasp is completed. The observation image size is $112 \times 112$ in PyBullet using a hand camera. More environment details can be found in Appendix Section 6.2.

**Training and Testing.** We use approximately $1,500$ ShapeNet objects from 169 different classes in the ACRONYM dataset [44] for training, where each object has 100 pre-computed grasps from [7] for planning. For testing, we use 9 selected YCB objects with 10 scenes per object and 138 holdout ShapeNet objects within 157 scenes. We train each model three times with different random seeds. We run each YCB scene 5 times and each ShapeNet scene 3 times and then compute the mean grasping success rate. More details on network architecture and training can be found in Appendix Section 6.3 and Appendix Section 6.4, respectively.

## 4.1 Ablation Studies in Simulation

**Ablations on State Representations.** We first conduct experiments with BC to investigate the effect of using different state representations. Table 1 presents the grasping success rates. When using images, there are three variations: "RGB", "RGBD" and "RGBDM" indicating whether depth (D) or foreground mask (M) of the object are used or not. Depth and/or mask are concatenated with the RGB image, and ResNet18 [45] is used for feature extraction. "Point" in Table 1 indicates the point cloud state representation. "Offline" means a fixed-size offline dataset $\mathcal{D}_{\text{expert}}$ is used for training which contains 50,000 data points from expert demonstrations. "Online" means the expert planner is running in parallel with training, which keeps adding new data to the dataset. "DAGGER" uses on-policy rollout data from the learner policy with expert supervision $\mathcal{D}_{\text{dagger}}$, and "Goal-Auxiliary" indicates whether we add the grasping goal prediction task or not in BC training.

From Table 1, we can see that: i) using the point clouds achieves better performance compared to using images of the current view. This indicates that the 3D features in object point clouds generalizes better for 6D grasping. ii) Adding depth or foreground mask to the image-based representation improves the performance. iii) "Online" is overall better than "Offline" by utilizing more data for training. iv) Both DAGGER and adding the auxiliary task improve success rates.

**Ablations on Goal-auxiliary vs. Goal-conditioned.** We evaluate our goal-auxiliary DDPG algorithm for 6D grasping, where the dataset $\mathcal{D} = \mathcal{D}_{\text{expert}} \cup \mathcal{D}_{\text{dagger}} \cup \mathcal{D}_{\text{ddpg}}$ is constructed for training. We also consider another common strategy of using goals for comparison: goal-conditioned policies such as in [30, 28, 31], where a goal is concatenated with the state as the input for the network. In this case, in order to use goals in testing, we need to train a separate network to predict goals from states. Table 2 displays the evaluation results, where we test different combinations of adding the goal-auxiliary task or using the goal-conditioned input to the actor and the critic.

| Input | Method | | | | Test | |
|---|---|---|---|---|---|---|
| | Offline | Online | DAGGER | Goal-Auxiliary | ShapeNet | YCB |
| RGB | ✓ | | | | 43.6 | 45.2 |
| RGB | | ✓ | | | 50.6 | 51.5 |
| RGB | | ✓ | ✓ | | 52.1 | 52.3 |
| RGB | | ✓ | ✓ | ✓ | 54.3 | 58.1 |
| RGBD | ✓ | | | | 45.3 | 46.3 |
| RGBD | | ✓ | | | 58.7 | 65.3 |
| RGBD | | ✓ | ✓ | | 60.4 | 67.1 |
| RGBD | | ✓ | ✓ | ✓ | 68.0 | 71.7 |
| RGBDM | ✓ | | | | 48.3 | 50.4 |
| RGBDM | ✓ | | | ✓ | 55.8 | 52.2 |
| RGBDM | | ✓ | | | 67.4 | 66.7 |
| RGBDM | | ✓ | ✓ | | 71.5 | 72.3 |
| RGBDM | | ✓ | ✓ | ✓ | 75.2 | 72.6 |
| Point | ✓ | | | | 73.6 | 65.6 |
| Point | ✓ | | | ✓ | 73.3 | 67.3 |
| Point | | ✓ | | | 72.7 | 72.1 |
| Point | | ✓ | | ✓ | 72.3 | 72.7 |
| Point | | ✓ | ✓ | | 75.8 | 77.2 |
| Point | | ✓ | ✓ | ✓ | **79.6** | **78.5** |

Table 1: Success rates of different models trained by behavior cloning with expert supervision.

| Policy / Critic | None | | Goal-Auxiliary | | Goal-Conditioned | | Both | |
|---|---|---|---|---|---|---|---|---|
| | ShapeNet | YCB | ShapeNet | YCB | ShapeNet | YCB | ShapeNet | YCB |
| None | 80.6 | 71.0 | 77.3 | 72.8 | 68.6 | 60.7 | 76.7 | 76.6 |
| Goal-Auxiliary | 84.6 | 82.8 | **91.3** | **88.2** | 83.4 | 81.3 | 87.8 | 84.1 |
| Goal-Conditioned | 81.6 | 73.7 | 79.3 | 76.1 | 70.3 | 63.2 | 82.1 | 77.9 |
| Both | 86.8 | 82.5 | 87.9 | 83.1 | 84.5 | 81.7 | 86.9 | 82.3 |

Table 2: Evaluation on success rates of two strategies of using goals in RL with DDPG.

From Table 2, we can see that: i) using the goal-auxiliary loss for the critic significantly improves the performance since it regularizes the Q-learning. ii) The goal-conditioned policies perform worse than the goal-auxiliary counterparts. When the predicted goal is not accurate, it affects the grasping success. We also observe some instability in goal-conditioned policy training. iii) Adding the goal-auxiliary task to both the actor and the critic achieves the best performance. iv) Comparing to the supervised BC training in Table 1, our GA-DDPG algorithm further improves the policy, especially in the contact-rich scenarios. Imitation learning alone cannot handle these contacts with the target object before closing the fingers since these are rarely seen in the expert demonstrations.

**Ablations on Fine-tuning with Hindsight Goals.** Our policy trained on ShapetNet objects achieves 91.3% and 88.2% success rate for grasping unseen ShapeNet and unseen YCB objects, respectively, as shown in Table 2. We can further improve the policy using RL with unseen objects. To resemble real-world RL, we assume that the 3D shape and pose of the YCB objects are not available. Therefore, there is no expert supervision on these objects. We fine-tune the policy using the dataset $\mathcal{D} = \mathcal{D}_{\text{expert}} \cup \mathcal{D}_{\text{hindsight}}$ with hindsight goals. We see that the success rate is improved from 88.2% to 93.5% after fine-tuning. We also experiment with the simulation dynamics by decreasing the lateral frictions of the object, which makes object easier to slide during contacts. In this case, the success rate of the pre-trained policy reduces to 81.6%. After fine-tuning with hindsight goals, the policy adapts well and achieves 88.5% success rate, which demonstrates the robustness of the training procedure to contact model errors.

**Ablations on Other Design Choices.** We conduct ablation studies on several design choices in GA-DDPG as shown in Table 3. We can see that: i) RL without BC failed to learn a useful policy due to the high-dimensional state-action space and sparse reward. ii) Both online interaction for adding data and DAGGER help. iii) Our aggregated point cloud representation is better than using "RGBDM" images or single-frame point clouds. iv) Early fusion of the state-action in the critic is better than late fusion, i.e., concatenating features of state and action. v) "Expert Init.": we also use expert-aided exploration by rolling out expert plans for a few steps to initiate the start states during learner rollouts, which is beneficial. vi) The point matching loss defined in Eq. (2) is better than a direct L2 loss on translation and rotation. Overall, we observe that these design choices both stabilize training and improve the performance. Ablation experiments on grasp prediction performance can be found in Appendix Section 6.5.

| Test | No BC | Offline | No DAGGER | Image | No Aggr. | Late Fusion | No Expert Init. | L2 Loss | Final |
|---|---|---|---|---|---|---|---|---|---|
| ShapeNet | 2.7 | 72.3 | 83.9 | 83.7 | 83.7 | 79.1 | 87.7 | 81.8 | **91.3** |
| YCB | 3.8 | 63.7 | 85.3 | 78.9 | 83.5 | 82.1 | 86.6 | 79.2 | **88.2** |

Table 3: Ablation studies on different components in our method with grasping success rates.

| | Fixed initial joint | | | Varying initial joint | | |
|---|---|---|---|---|---|---|
| Object | Policy | Planning | Combined | Policy | Planning | Combined |
| YCB | 35/45 | 38/45 | 37/45 | 31/45 | 26/45 | 34/45 |
| Unseen | 8/10 | 5/10 | 8/10 | 7/10 | 7/10 | 7/10 |
| Success Rate | 78.2 | 78.2 | **81.8** | 69.1 | 60.0 | **74.5** |

Table 4: Real-world grasping results for single objects on a table. The numbers of successful grasps among trials and the overall success rates are presented.

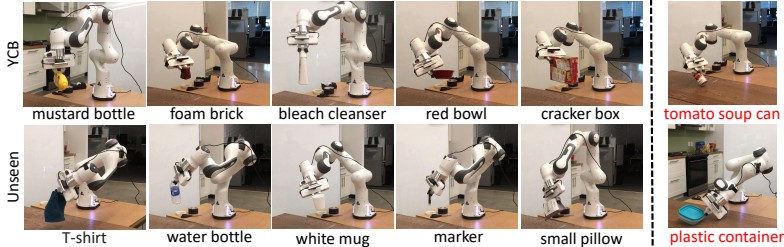

Figure 3: Successes (left side) and failures (right side) of real-world tabletop 6D grasping using our policy trained in simulation.

## 4.2 Tabletop 6D Grasping in the Real World

We conducted tabletop grasping experiments on a Franka robot with a realsense D415 camera on the robot gripper. We evaluated two different settings: fixed initial setting and varying initial setting. In the first setting, the robot always starts from the same initial position but the object positions change to evaluate the robustness of GA-DDPG with respect to different object poses. In the second setting, we evaluate the robustness of the system against different intial positions of the robot. For each setting we consider 3 baselines: a) Policy: we use GA-DDPG during the whole trajectory and grasping the object. b) Planning: we use GraspNet [10] to generate grasps for the object and OMG planner [11] for planning. GraspNet trains a variational autoencoder to generate 6D grasps from an input point cloud of an object, while the OMG planner is a trajectory optimization method for joint motion planning and grasp selection. c) Combined: The planner plans trajectories with length of 25 steps. We execute the first 20 steps of the plan and switch to GA-DDPG for the final part. Objects are segmented using unseen object instance segmentation method [46] for all the settings. We evaluated different variations on 9 YCB objects and 10 unseen objects with varying shapes and materials. GA-DDPG is finetuned on YCB objects. For each YCB object, we tested 5 different configurations according to the fixed/varying initial settings. Table 4 presents the numbers of successful grasps among trials. Fig. 9 in the Appendix shows all the objects in our experiments.

We can see that the learned policy performs on par with the open-loop planning with fixed initial robot poses and outperforms it in the varying initial setting. Combining planning with the policy achieves the best performance in both settings. For varying initial poses, open-loop planning suffers more due to perception errors and control errors, since the robot started further away from the target object. Using the policy for the last steps of grasping can fix some open-loop failures in the combined version. However, since the policy is trained in simulation, it suffers from the sim-to-real gap, especially in contact modeling with objects. Improving the fidelity of physics simulation and also fine-tuning with real-world data would help bridge the gap. A discussion on design limitations can be found in Appendix Section 6.8. Fig. 3 shows some successful and failed grasping examples. Most failures in the real world are due to the sim-to-real gap in contacting modeling.

## 4.3 Human-to-Robot Handovers in the Real World

Our closed-loop policy can be applied to human-to-robot (H2R) handovers. Based on the perception system introduced in [47], we integrate GA-DDPG to compute actions for robot control. Specifically, the point cloud of an object in hand was segmented using an external Azure Kinect RGB-D camera according to [47]. Then the object point cloud is fed into the policy network to compute the control action. As shown in Fig. 4 (a), we used a subset of 10 household objects proposed in [47], and measured the *time to successful handovers*, *success rate* and the *number of grasp attempts*. We first conducted a systematic evaluation to measure the performance of our approach with respect to different objects and varied handover locations. Then we conducted a user study with 6 participants recruited from the lab. We asked the participants to answer a questionnaire with Likert-scale questions and open-ended questions after the experiments.

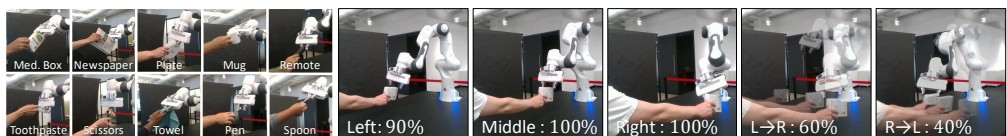

Figure 4: (a) The 10 unseen household objects we used for the human-to-robot handover experiments. *Please zoom in for the best view.* (b) The systematic evaluation with varied handover locations (left, middle, right) and moving objects during the handover (from left to right, and from right to left). We report the success rate for each setting. The overall success rate is 78%.

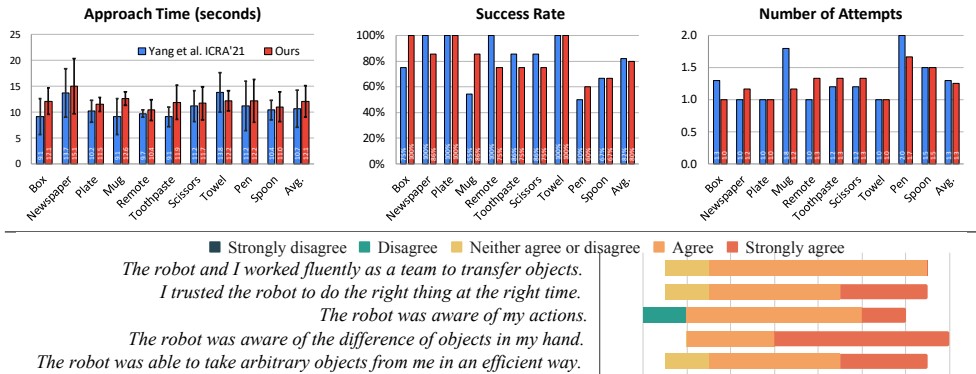

Figure 5: Top: Quantitative results of the user study with 6 participants and 10 household objects. Bottom: Participants' agreement with each statement in the questionnaire.

**Systematic Evaluation.** We handover each object from three handover locations, i.e., *left, middle*, and *right*, with respect to the robot base location. Then we tried to move the object from left to right, and from right to left, after the robot starts to move. We conducted experiments on all the 10 household objects, and report the success rate in Fig. 4 (b) with an example of *Medicine Box* on these 5 settings. Please refer to Table 7 in Appendix for detailed results. Overall, reactive handover with moving objects is more challenging than static handover. The robot sometimes cannot adjust its grasping after the object moves. In our setting, the camera is on the right side. When the object is on the left side, i.e., further from the camera, we see performance drop due to point cloud sensing noises, especially for the right to left reactive handover. Additionally, the Azure Kinect only runs at 15fps, which also causes delays in reactive handover.

**User Study Results.** As reported in Fig. 5 Top, our method achieves comparable performance against [47] which is specifically designed and tuned for the task of H2R handovers. Interestingly, our approach has higher success rate for some difficult objects, such as *mug* and *pen*, which demonstrates that our approach can not only generalize to moving objects, but also adapts well to unseen environments (non table-top settings) and views (external camera).

**Subjective Evaluation.** Fig. 5 Bottom reports participants' agreement with statements from [47]. Overall, the participants agreed with the fluency and commented "[the robot] fluently moved toward the object" and "could adjust it's position when I moved". They felt safe, saying "[I] never felt it would overshoot or become aggressive". They also sensed the robot was aware of their actions since "[it] adapted fast when I moved my arm", and was aware of the difference of objects "because it used different grasps for different objects". One was "surprised it worked on very thin objects". Several participants thought the robot motion was slow when it got close to the object. This is because the system evaluates the grasping score using the grasp evaluator in GraspNet [10]. Only when the score is higher than a pre-defined threshold, the robot closes its gripper.

## 5   Conclusion

We introduce the goal-auxiliary DDPG algorithm for efficiently learning of 6D grasping control polices from point clouds. Our method uses demonstrations from an expert motion and grasp planner and utilizes grasping goal prediction as an auxiliary task to improve the performance of the actor and the critic. We demonstrate that our policy trained in simulation can be integrated in a tabletop 6D grasping system and a human-to-robot handover system to improve grasping performance of unseen objects. For future work, we plan to investigate the sim-to-real gap in policy learning due to contact modeling in simulation and extend the method to 6D grasping in cluttered scenes.

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
