# OpenReview forum: "Goal-Auxiliary Actor-Critic for 6D Robotic Grasping with Point Clouds"
_robot-learning.org/CoRL/2021/Conference — CoRL2021 Poster_

### Official Review · Reviewer_GYJh · 2021-07-05

**Originality:** Good
**Technical Quality:** Very Good
**Clarity Of Presentation:** Very Good
**Impact:** 4

**Recommendation:**

Weak Accept: I recommend accepting the paper, but will not argue for my recommendation if the majority of other reviewers have a different opinion.

**Summary:**

The paper proposes a method for learning closed-loop 6D robot grasping policies of singulated objects, using segmented aggregated point clouds as input. It proposes a combination of behavior cloning from expert demonstrations and reinforcement learning based on an actor-critic approach. Furthermore, it introduces an auxiliary task for supervised training of the actor and critic networks, which consist of predicting the final grasping pose of the end-effector. During the behavior cloning phase, additional demonstrations are gathered following the DAGGER scheme, where the expert provides the labels for actions and goals. During the RL phase, goal labels are generated either from a set of pre-labeled goals (in case of known objects with annotations) or hind-sight goals are used (in case of unknown objects) to add a goal label to the data tuples in the replay buffer.
Extensive ablation studies are conducted in simulation on the algorithmic choices and on different input representations underlining their benefit. The paper also show-cases the method on two real world tasks: Table-top grasping and human-to-robot handover.


**Issues:**

- The method for aggregating point clouds is not mentioned in the methodic section of the main paper. The first place where the word aggregation appears (but not explained) is in the section 'Ablation on Other Design Choices'. Since it is however an integral part of the method proposed in the paper, it should be mentioned earlier, at least with a reference to the appendix.

- In lines 212 - 219: The paper states: 'From table 2 we can see that ... iv) Comparing to the supervised BC training, our GA-DDPG algorithm further improves the policy, ...' There seems to be no mentioning of BC in that table. Can the authors clarify, what they refer to?

- In line the authors state 214-215: ii) 'The goal-conditioned policies perform worse than the goal-auxiliary counterparts. When the predicted goal is not accurate, it affects the grasping success.' Did the authors investigate that further, or is it just a hypothesis?

- In line 226 the paper states: 'We see that the success rate is improved from 88.2% to 93.5% after fine-tuning.'  Where can we see that? It would be nice to add this result to table 3.

-  A discussion of the sensitivity of the proposed method wrt. hyperparameters (e.g. the weighting lambda of the actor rewards ) would be interesting

**Reviewer Expertise:**

Very good: Comprehensive knowledge of the area

**Strengths And Weaknesses:**

Strength:
- The paper contains extensive ablation studies on the different parts of the method, which do not only give insight into the method proposed in the paper, but can also give useful hints to other research in the field.
- The combination of imitation learning and reinforcement learning is promising for an increased sample efficiency of robotics manipulation tasks, in particular in the proposed combination with DAGGER (-like) training and hind-sight goal labelling.
- The idea of using goal prediction as auxiliary task is interesting and seems to lead to surprising performance gains as compared to goal conditioning.

Weaknesses:
- Many important technical points were transferred to the appendix, without reference, but are actually crucial for an understanding of the proposed method,e.g. point cloud aggregation, the training of the goal prediction network. There should at least be a to the relevant section in the appendix, but a discussion in the main paper would be better.
- There seem to be missing some numbers in the tables, which are referenced in the text. In particular table 2 and table 3.
- The benefit of 6D grasping does not become obvious to me from the experiments on grasping. A comparison of a method capable of top-down grasps only would be useful.

**Summary Of Recommendation:**

I liked reading the paper and found  the thorough ablation studies very insightful. In particular the comparison of different input modalities is interesting and can provide very useful insights to the community. The proposed methods for to both closed loop grasping and 6D grasping constitute relevant contribution to research in that field.
I therefore recommend the acceptance of the paper.

---

> ### Author Response · Authors · 2021-08-21
> **Responses to Reviewer GYJh**
>
> Thank you for your comments and suggestions. We have updated our paper and will address specific questions below.
>
> Q1: Many important technical points were transferred to the appendix
>
> Thanks for pointing this out. We will refer to these technique points in the main paper.
>
> Q2: There seem to be missing some numbers in table 2 and table 3
>
> We will fix this.
>
> Q3: Benefit of 6D grasping from the experiments, comparison of top-down grasping
>
> Top-down grasping methods cannot handle all scenarios in our experiments, especially in human-to-robot handover. A simple example is, if the object is a bit far from the robot, the robot cannot reach the top-down position. In this case, 6D grasping methods can grasp from sides of the object.
>
> Q4: The method for aggregating point clouds is not mentioned in the methodic section
>
> We will fix this. Since point cloud aggregation is only used for grasping static objects. We put this section in the appendix. We will make this clear in the paper.
>
> Q5: In lines 212 - 219: no BC mentioned in table 2
>
> The BC results are presented in table 1 with different representations. The last row of table 1 is a baseline BC algorithm compared to RL. We will make this clear.
>
> Q6: Did the authors investigate that goal-conditioned policies perform worse due to goal prediction errors?
>
> This is supported by the ablation study on how to use goals in Table 2. A goal-conditioned policy takes a state and a goal as input , so the grasping performance is affected if the goal is not accurate. We also observed in earlier experiments that the policy tries to reach bad goals and gets stuck in contacts.
>
> Q7: In line 226: the success rate is improved from 88.2% to 93.5% after fine-tuning. Where can we see that?
>
> Since there is only one experiment on fine-tuning with hindsight goals for YCB objects, we did not add this number to Table 3.
>
> Q8: Sensitivity of the proposed method wrt. hyperparameters (e.g. the weighting lambda of the actor rewards)
>
> We agree. The weighting lambda needs to be in a valid range to avoid divergence and improve performance. We did not do a very extensive parameter search for this hyperparameter.

---

> > ### Comment · Reviewer_GYJh · 2021-08-27
> > **Response to authors**
> >
> > Dear Authors,
> > thank you for your explanations and clarifications.
> > Concerning Q3:
> > While top-down grasping might not be enough for the human hand-over task, it would still be interesting to see a comparison to top-down grasping for the other experiments. I do understand, however, that this is not the focus of the current paper and do therefore not see this comparison as a necessity.

---

> ### Comment · Reviewer_GYJh · 2021-09-01
> **Revised Review**
>
> The authors addressed most raised issues in their answers and revised paper.
> I still somewhat miss a comparison to top-down grasping methods in the experiments, where top-down grasping is possible, even thouhg I understand that this is not the main focus of the paper.
> I therefore do not change my initial rating and still reommend weak accept.

---

### Official Review · Reviewer_nhX8 · 2021-07-20

**Originality:** Good
**Technical Quality:** Very Good
**Clarity Of Presentation:** Very Good
**Impact:** 3

**Recommendation:**

Weak Accept: I recommend accepting the paper, but will not argue for my recommendation if the majority of other reviewers have a different opinion.

**Summary:**

The paper presents a system for learning-based 6D Grasping. The method combines imitation and reinforcement learning: use imitation learning to overcome the initial exploration challenges, and use reinforcement learning to further improve the policy performance, especially on shapes that are hard to create expert demonstration for. The method is evaluated on grasping a variety of objects in both simulated and real environments, as well as a human-robot handover task.

**Issues:**

See main comments.

**Reviewer Expertise:**

Very good: Comprehensive knowledge of the area

**Strengths And Weaknesses:**

I like the practicality of the system: it makes intuitive sense to combine imitation learning and reinforcement learning for challenging problems like learning reactive grasping. The fact that the system operates on partial point cloud observation makes sim-to-real transfer feasible. I also appreciate the final human-robot handover experiment that shows why reactive grasping is important.

However, I found it difficult to distill the thesis of the paper: what should the readers learn from the paper beyond knowing that the system “works”? From a technical novelty perspective, the key contributions seem to be spread across a few points: 1) combining imitation learning and reinforcement 2) learning from point cloud data for 6 DoF grasping and 3) the goal auxiliary loss.

Like mentioned in the paper, combining imitation and reinforcement learning is not a new idea in visual-motor learning, and the idea of learning from raw point cloud for grasping has been explored by a few prior works too. The goal auxiliary losses seems new, but I’m not entirely sure what’s the motivation behind this change. What challenges are these loss functions designed to address? The only place that discusses this point is 149-151: “We introduce two auxiliary tasks for the actor-critic to predict goals, which help to learn the policy and the Q-function by adding goal supervision”, but it doesn’t really motivate *why* these losses are added. I wish the authors could shed more light on the key takeaways of the paper in the next iteration.

On the other hand, the paper does fulfill its high-level goal of developing a learning-based reactive grasping system. And the system seems to fit especially well for human-robot handover tasks judging by the experiment evaluation. However, I had a similar struggle to understand the key differences between the proposed system and the prior works. The only relevant point made in the paper is that “it performs on-par with the prior work”. What are the assumptions relaxed, if any, compared to the prior work that made this result significant? I’d like to see a more in depth discussion on the result.

Minor:
What’s the input space for real-world experiments? I don’t see it mentioned anywhere in 4.2 and 4.3.


**Summary Of Recommendation:**

Like mentioned in the comment, I like the practicality of the system and appreciate the evaluation. But I struggled to distill the key lessons from this paper. Hence before seeing more comments from the author, I do not recommend accepting the paper.

---

> ### Author Response · Authors · 2021-08-21
> **Responses to Reviewer nhX8**
>
> Thank you for your comments and suggestions. We have updated our paper and will address specific questions below.
>
> Q1: What should the readers learn from the paper beyond knowing that the system “works”?
>
> We believe our work brings significant insights to the robotic grasping field, especially on how to learn closed-loop control policies for 6D grapsing. Although several ideas have been introduced in the RL field such as combining IL and RL, these ideas have not been applied to 6D robotic grasping. We show that our design of the system enables us to learn closed-loop policies for 6D grasping that work in the real world.
>
> Instead of just showing the system works, we also show why the system works. Our ablation studies investigate state representation, policy learning with BC and RL, different ways of using goal information, and several other design choices. We believe these analyses are valuable to readers in the robotic grasping field.
>
> Q2: What is the motivation behind the goal-auxiliary losses? What challenges are these loss functions designed to address?
>
> Auxiliary losses are usually introduced to facilitate learning. We can find many examples of using auxiliary losses in the machine learning literature and goal usage in the RL literature.
>
> In our case, adding the goal-auxiliary losses stabilizes training and improves the learned policy. Adding goal prediction forces the network to learn where to grasp. In table 3, we can clearly see that after adding the goal-auxiliary losses, the grasping success rate is significantly improved (80.6 to 91.3 for ShapeNet objects and 70.1 to 88.2 for YCB objects).
>
> Q3: Struggle to understand the key differences between the proposed system and prior works
>
> The main difference is that the prior works are open-loop grasping. They combine grasp planning and motion planning from partial point clouds for 6D grasping. Ours is a learned closed-loop control policy for 6D grasping.
>
> Our system performs on par with the open-loop baseline in the fixed initial joint settings in Table 4, but significantly outperforms it in the varying initial joint settings that are more challenging.
>
> Q4: What is the input space for real-world experiments?
>
> In the real world, the input to our policy is a segmented point cloud of an object.

---

> > ### Comment · Reviewer_nhX8 · 2021-09-03
> > **Thanks for the reply**
> >
> > I appreciate the reply. While the response has partially addressed my concern about the novelty of this paper, I'm still not fully convinced that the paper offers significant technical insight. But again, I do appreciate the practicality of the system. Therefore I have changed my rating to weak accept.

---

### Official Review · Reviewer_i4Zo · 2021-07-23

**Originality:** Good
**Technical Quality:** Very Good
**Clarity Of Presentation:** Very Good
**Impact:** 3

**Recommendation:**

Weak Accept: I recommend accepting the paper, but will not argue for my recommendation if the majority of other reviewers have a different opinion.

**Summary:**

The paper proposes a method for closed-loop 6D robotic grasping which learns a policy that maps a state corresponding to a point-cloud of the scene to an action that corresponds to 6D transformation of the gripper pose. The learned model uses an actor-critic schema. The actor is trained using Behavior Cloning approximating the trajectories generated by a motion planner expert, and the critic is trained to predict the expected return of the trajectory that correspond to the grasp probability. Both modules also predicts a goal pose as an auxiliary task. The method is evaluated on Shapenet and YCB object in simulation and in real world, demonstrating nice results in static setting (grasping from table top) as well as dynamic setting (human to robot handover). The use of different modalities from inference as well as different learning features are also ablated in the experimental section.

**Issues:**

Nothing beyond what was mentioned above.

**Reviewer Expertise:**

Very good: Comprehensive knowledge of the area

**Strengths And Weaknesses:**

The method in general makes sense, and the performance seems to be quite good. The results demonstrate the value of the method mostly in dynamic mode where closed-loop is actually required, and to some extent also in static scenes there is some benefit compare to an open-loop prior solution.
I believe that a comparison to other closed-loop 6D methods such as "Grasping in the Wild: Learning 6DoF Closed-Loop Grasping from Low-Cost Demonstrations" could improve the quality of the paper.
In addition, a few references (e.g. OMG, Graspnet) are used either as sub-modules of the proposed approach or for comparison, which has some commonality but also some difference. It could help the reader to have some short description of each reference to understand the deviations.

**Summary Of Recommendation:**

Overall the paper is quite nice.

---

> ### Author Response · Authors · 2021-08-21
> **Responses to Reviewer i4Zo**
>
> Thank you for your comments and suggestions. We have updated our paper and will address specific questions below.
>
> Q1: a comparison to other closed-loop 6D methods such as “Grasping in the Wild” [22] could improve the quality of the paper
>
> We agree with the reviewer. However, [22] is not open-sourced. In addition, their evaluation settings are similar to top-down grasping and a fair comparison would require new data collection.
>
> Q2: Have some short descriptions of each sub-module reference
>
> We will add descriptions for submodules and comparison methods such as 6D GraspNet and OMG Planner.

---

### Official Review · Reviewer_7sEr · 2021-07-23

**Originality:** Good
**Technical Quality:** Excellent
**Clarity Of Presentation:** Very Good
**Impact:** 4

**Recommendation:**

Strong Accept: I recommend accepting the paper and will argue for my recommendation even if other reviewers hold a different opinion.

**Summary:**

This work presents an approach for learning closed-look policies for 6D grasping arbitrary objects. Given an RGBD input, the following is done:
(1) The object is  segmented from the environment using the RGB input.
(2) The object's segmentation mask is used to get the point cloud corresponding to the object (assuming a calibrated camera, so that the depth image can be used to infer this point cloud).
(3) The point cloud is used an input to two PointNet++ networks, which act as the actor and critic.
(4) The action from the actor (a 6D offset from the current gripper pose) from the actor is executed.
(5) Once a goal grip position is reached, an attempt at grasping the object is made.

The key novel idea in the above is to add an auxiliary objective to both the actor and critic, which is to predict the final grasp pose. This can be trained via supervised learning, making the task of learning good actions and value estimates easier, since these are informed by what a good final grasp pose is.

Learning is done in via a combination of behavioral cloning and deep RL in two steps:
(1) A dataset of expert trajectories is generated by a motion planner in simulation, with grasp planning utilized to select the end positions for the motion planner. DAGGER-based behavior cloning is further used to get a more robust set of data.
(2) The above dataset is included in the replay buffer and RL is done with additional losses for BC and the auxiliary objective. A sparse reward of successful grasping is used and TD3 is used as the RL algorithm.

Detailed evaluation and ablation experiments are done in simulation, demonstrating high (~90%) success rates and the need for all the aspects of this approach convincingly. Additional evaluation is done in a real world setting, in which a lower success rate (~70) is seen due to the sim-to-real gap. It is also demonstrated that this approach can be used to implement human-to-robot handovers of objects.

**Issues:**

"Expert Init" is listed in Table 3, but I can't find it explained prior to this.
More details regarding failure modes in both simulation and real would be useful, as well as more discussion of what the sim-to-real gap is in this case.

**Reviewer Expertise:**

Excellent: Expert knowledge on the topic of the paper

**Strengths And Weaknesses:**

Strengths:
* Closed loop 6D grasping of arbitrary objects given raw sensor information is an important problem in robotics that is far from solved
* The paper is well written, with no significant ambiguities
* The problem formulation for training makes no assumptions about prior knowledge about the environment, and while some knowledge of object models is used the RL training can also proceed without that
* Using point clouds as inputs is a good idea, as it allows for zero-shot transfer from simulation to real with a lesser discrepancy in sensing than would be the case with RGB inputs
* The concept of the auxiliary loss is novel and fitting for this problem
* The evaluation is very well done, with an extremely well done ablation study and impressive results. The additional demonstration of object hand-over results also further demonstrates this approach's value.
* The supplementary material contains additional valuable information, in particular more detail concerning the hindsight goal training and details about the training process and informative learning curves

Weaknesses:
* The approach is primarily composed of ideas introduced in prior works. The auxiliary loss being the main novelty and is interesting, but has limited applicability in other settings.
* The method relies on having a robust segmentation algorithm for arbitrary objects
* The method requires learning in simulation, as it needs a large number of interaction steps to converge
* The performance suffers in the real world due to the sim-to-real gap due to discrepancies in physics simulation (in particular contact modeling)
* Also the approach compatible with grasping in clutter, this is not attempted in this work.
* Fine-tuning in the real world is suggested as a possible solution to the sim-to-real gap, but is not attempted in this work

Several of the weaknesses indicate directions for future work, rather than significant issues with this paper.

**Summary Of Recommendation:**

This paper is very well done and presents a well thought out training approach for an important and as of yet rarely addressed problem. The main weakness is having few original ideas, but the strength of how different ideas are combined and the value of the auxilliary loss make this a relatively small issue.

---

> ### Author Response · Authors · 2021-08-21
> **Responses to Reviewer 7sEr**
>
> Thank you for your comments and suggestions. We have updated our paper and will address the weaknesses mentioned by the reviewer below.
>
> Q1: The approach is primarily composed of ideas introduced in prior works. The auxiliary loss has limited applicability in other settings.
>
> We agree with the reviewer that several ideas in our work have been introduced in previous RL literature. Our goal is to apply these ideas to tackle closed-loop 6D grasping and we introduce the goal-auxiliary loss to improve policy learning. The specific loss function cannot be used for other settings, but the idea of using goal-auxiliary loss can be applied.
>
> Q2: The method relies on having a robust segmentation algorithm for arbitrary objects.
>
> Our policy is robust to noises in the segmentation. We see cases where only a part of an object is segmented, but the policy can still grasp the object on the segmented part.
>
> Q3: The method requires learning in simulation, as it needs a large number of interaction steps to converge.
>
> RL in the real world is challenging and time-consuming. Learning in simulation is a good way to bootstrap. Combining IL and RL significantly reduces the number of interactions needed in our work.
>
> Q4: Sim-to-real gap due to discrepancies in physics simulation, and future work on grasping in the clutter and real-world fine-tuning
>
> Since we learn the policy in simulation, our method indeed suffers from the sim-to-real gap in the physics simulation. One idea to solve this is to perform fine-tuning of the policy in the real world. We will work on this and extend the method to 6D grasping in clutter.
>
> Q5: Definition of "Expert Init" in Table 3
>
> The definition of “Expert Init” is in lines 236-238. We will make this clear.
>
> Q6: More details and discussions regarding failure modes in both simulation and the real world
>
> We will add more discussion on failure modes in the paper. Some discussions can be found in appendix 6.8. Most failures in the real world are due to the sim-to-real gap in contacting modeling.

---

> > ### Comment · Reviewer_7sEr · 2021-08-31
> > **Response**
> >
> > Thank you to the authors for their response. I think they address my concerns well.

---

### Meta-Review · Area_Chair_xJYD · 2021-08-13

**Recommendation:** Accept (Poster)
**Confidence:** 4

**Metareview:**

The reviewers have agreed that the paper has been addressing an important problem in robotic, i.e. closed loop 6D grasping of arbitrary objects. The combination of imitation learning and reinforcement learning is sensible, and the concept of the auxiliary loss is novel and fitting for this problem. However there is a suggestion to elaborate with motivations why these losses are added. In overall, though the idea is incremental, i.e. the combination of imitation learning and reinforcement learning, the evaluation is very well done, with a well done ablation study and impressive results. There are concerns that the key differences between the proposed system and the prior works  and under what assumptions (e.g. robust segmentation algorithm, grasp a single object) are unclear. There are also suggestions that a comparison of a method capable of top-down grasps or other closed-loop grasping methods should be included.

During the rebuttal phase, the authors have addressed most concerns by the reviewers. Though there are still open issues, e.g. comparisons to existing 6d close-loop and open-loop table-top grasping methods, the authors have demonstrated the efficiency and practicality aspects of their approach through many interesting evaluations.

---

> ### Author Response · Authors · 2021-08-21
> **Responses to Meta-Review**
>
> We thank all the reviewers and the area chair for their valuable comments. We have revised the paper as suggested by the reviewers. Overall, most reviewers agree with the contributions of our paper. Reviewer nhX8 seems to have some concerns about the contributions and novelty of the paper. We emphasize that the main contribution is a robust 6D reactive grasping system.
>
> We address specific questions in the meta review below.
>
> Q1: Elaborate with motivations why these auxiliary losses are added
>
> We will elaborate motivations of using the auxiliary losses in the paper. Overall, adding the auxiliary losses stabilizes training and improves the learned policy.
>
> Q2: What are the key differences between the proposed system and the prior works?
>
> The main difference is that we learn a closed-loop 6D grasping policy that can be used for either table-top grasping or human-to-robot handover. Previous works for 6D grasping are either open-loop or focus on bin-picking scenarios.
>
> Q3: Under what assumptions (e.g. robust segmentation algorithm, grasp a single object) are unclear.
>
> We will clarify the assumptions in the paper. The segmentation does not need to be very robust for our policy. We see cases where only a part of an object is segmented, but the policy can still grasp the object on the segmented part.
>
> Our policy is trained for grasping single objects. To deal with cluttered scenes, we could combine the policy with a motion planning algorithm for obstacle avoidance. We consider this as future work.
>
> Q4: A comparison of a method capable of top-down grasps or other closed-loop grasping methods should be included
>
> Top-down grasping methods cannot handle all scenarios in our experiments, especially in human-to-robot handover. Previous closed-loop grasping methods are either top-down grasping or not open-sourced for comparison such as [22]. We will release our code for future comparison.

---

### Decision · Program_Chairs · 2021-09-13

**Decision:**

Accept (Poster)

**Comment:**

The reviewers have agreed that the paper has been addressing an important problem in robotic, i.e. closed loop 6D grasping of arbitrary objects. The combination of imitation learning and reinforcement learning is sensible, and the concept of the auxiliary loss is novel and fitting for this problem. However there is a suggestion to elaborate with motivations why these losses are added. In overall, though the idea is incremental, i.e. the combination of imitation learning and reinforcement learning, the evaluation is very well done, with a well done ablation study and impressive results. There are concerns that the key differences between the proposed system and the prior works  and under what assumptions (e.g. robust segmentation algorithm, grasp a single object) are unclear. There are also suggestions that a comparison of a method capable of top-down grasps or other closed-loop grasping methods should be included.

During the rebuttal phase, the authors have addressed most concerns by the reviewers. Though there are still open issues, e.g. comparisons to existing 6d close-loop and open-loop table-top grasping methods, the authors have demonstrated the efficiency and practicality aspects of their approach through many interesting evaluations.